# A Multiscale Numerical Modeling Study of Smoke Dispersion and the Ventilation Index in Southwestern Colorado

**Michael T. Kiefer** [1,*], **Joseph J. Charney** [2], **Shiyuan Zhong** [1], **Warren E. Heilman** [2], **Xindi Bian** [2] **and Timothy O. Mathewson** [3]

1   Department of Geography, Environment, and Spatial Sciences, Michigan State University, Geography Building, 673 Auditorium Rd., Room 116, East Lansing, MI 48824, USA; zhongs@msu.edu
2   Northern Research Station, USDA Forest Service, 3101 Technology Blvd., Suite F, Lansing, MI 48910, USA; joseph.j.charney@usda.gov (J.J.C.); warren.heilman@usda.gov (W.E.H.); xindi.bian@usda.gov (X.B.)
3   U.S. Department of the Interior, Bureau of Land Management, 2850 Youngfield Street, Lakewood, CO 80215, USA; t2mathew@blm.gov
*   Correspondence: mtkiefer@msu.edu

**Abstract:** The ventilation index (VI) is an index that describes the potential for smoke or other pollutants to disperse from a source. In this study, a Lagrangian particle dispersion model was utilized to examine smoke dispersion and the diagnostic value of VI during a September 2018 prescribed fire in southwestern Colorado. Smoke dispersion in the vicinity of the fire was simulated using the FLEXPART-WRF particle dispersion model, driven by meteorological outputs from Advanced Regional Prediction System (ARPS) simulations of the background (non-fire) conditions. Two research questions are posed: (1) Is a horizontal grid spacing of 4 km comparable to the finest grid spacing currently used in operational weather models and sufficient to capture the spatiotemporal variability in wind and planetary boundary layer (PBL) structure during the fire? (2) What is the relationship between VI and smoke dispersion during the prescribed fire event, as measured by particle residence time within a given horizontal or vertical distance from each particle's release point? The ARPS no-fire simulations are shown to generally reproduce the observed variability in weather variables, with greatest fidelity to observations found with horizontal grid spacing of approximately 1 km or less. It is noted that there are considerable differences in particle residence time (i.e., dispersion) at different elevations, with VI exhibiting greater diagnostic value in the southern half of the domain, farthest from the higher terrain across the north. VI diagnostic value is also found to vary temporally, with diagnostic value greatest during the mid-morning to mid-afternoon period, and lowest during thunderstorm outflow passage in the late afternoon. Results from this study are expected to help guide the application of VI in complex terrain, and possibly inform development of new dispersion potential metrics.

**Keywords:** smoke dispersion; ventilation index; complex terrain; numerical model; thunderstorm outflow

## 1. Introduction

In this study, FLEXPART-WRF [1], a Lagrangian particle dispersion model, was employed to examine the diagnostic value of the ventilation index (VI) for smoke dispersion in complex terrain in southwestern Colorado. The VI is most commonly computed as the product of the mixing height and the mean wind speed in the mixed layer, also referred to as the transport wind speed. The VI, often in the form of a ventilation adjective rating (VAR) (e.g., "Poor", "Good"), is a standard part of fire weather forecasts issued by National Weather Service (NWS) offices across the United States (US),

and the VAR is used by fire managers to assess the potential for dispersion of smoke from prescribed fires. In states across the US, including Colorado, state regulations specifically cite VI (also known as clearing index) thresholds for determining whether conditions satisfy legal authorization criteria for burning [2–4]. However, VAR category thresholds can vary considerably from state to state and from one NWS forecast office to another, with identical VI value ranges associated with a "Poor" VAR in one location and a "Good" VAR in another [5]. Since VI is used by fire managers across the US, including in areas of complex terrain [6], a study of the relationship, if any, between VI and smoke dispersion in and around complex terrain is potentially of great value. Drawing from anecdotal sources, [5] warn that VI is incapable of accounting for local flows in complex terrain and does not take into account the influence of complex terrain on mixing heights. However, to the best of the authors' knowledge, only two studies have examined VI and its diagnostic utility for smoke dispersion, in complex terrain or otherwise: [7,8].

The authors of [7] compared particulate matter (PM) concentrations and VI computed from on-site weather data collected during the Southeastern Smoke Project, a prescribed fire experiment in the Sandhills region of South Carolina. The study found little correspondence of PM concentration and VI, although [8,9] have pointed out that the relationship might have been stronger had the comparison been made farther away from the burn site. The authors of [8] used FLEXPART-WRF to simulate PM dispersion in and around a water gap in a ridgeline near the eastern edge of the Appalachian Mountains. The location and date of the study (Lehigh Gap, Slatington, PA, USA; 14 April 2013) were selected to coincide with a prescribed fire event, ensuring that the simulated weather conditions driving the smoke dispersion model were conditions in which prescribed fires in eastern Pennsylvania are conducted. Using a methodology similar to the one used in the current study, The authors of [8] found that VI had some diagnostic value for smoke dispersion potential in complex terrain, but that the relationship of dispersion and VI was sensitive to the position relative to the terrain features. VI was shown to have greatest diagnostic value upwind of terrain obstructions; however, the index failed as a diagnostic tool for dispersion potential in the lee of terrain obstructions, and in the vicinity of gaps or passes. In the latter areas, VI was unable to capture local terrain-induced flows because of its reliance on mixing height and transport wind speed, both of which exhibit limited sensitivity to underlying terrain.

As stated by [8], the model assessment of VI in the Lehigh Gap is best thought of not as a conclusive assessment of VI in complex terrain, but as a point of departure for additional studies of smoke dispersion and the diagnostic value of VI in complex terrain. A methodology was developed in that study that allows for a quantitative as well as qualitative assessment of VI as a diagnostic tool for smoke dispersion. A strategy was developed to release a large number of particles uniformly across an area containing a variety of topographic features (e.g., ridgeline and river gap) and terrain-mean flow configurations (e.g., windward and leeward slopes). VI and PM dispersion were analyzed for hypothetical release zones constructed by isolating clusters of particles released within different areas of the model domain. Five such hypothetical release zones (termed "analysis zones") were chosen to represent areas of the domain with different topographic characteristics. The authors devised the concept of a particle residence time (RT) within a given radius or depth as a quantitative measure of particle dispersion in the horizontal and vertical dimensions, respectively. They subsequently evaluated the statistical relationship of particle RT (i.e., dispersion) and VI for each analysis zone. Three-dimensional scatter plots of particle position allowed the authors to also qualitatively compare particle dispersion from the analysis zones. In [8], as in this study, diagnostic value is indicated by the coefficient of determination and the slope of a linear fit between VI and particle RT.

Limitations of [8] include the limited terrain complexity of the Lehigh Gap area and the occurrence of a consistently well-mixed atmosphere and narrow range of VI values during the event. The current study is an application of the method developed in [8] to a location with greater terrain complexity and an event with a wider range of VI values. The focus of the present study is a prescribed fire conducted from 8–12 September 2018 in the Saul's Creek area of the San Juan National Forest in southwestern

Colorado (hereafter referred to as the "Saul's Creek fire"). The Saul's Creek fire was part of a series of prescribed fires conducted on the Columbine Ranger District, about 30 km east of Durango, Colorado, with about 2000 ha of Ponderosa Pine, Gambel Oak, and grass understory burned in total. With a specific goal of evaluating VAR category thresholds in southwestern Colorado, a suite of observations were collected during the Saul's Creek fire in order to characterize the background meteorological conditions likely impacting smoke plume evolution (e.g., wind profiles from a portable rawinsonde system; Section 2.1). The selection of the Saul's Creek fire for the next stage in the evaluation of VI diagnostic value is based on both the setting of the fire in a region of complex terrain, with surface elevation changes on the order of 1 km over a distance of 30 km, and the availability of observational data with which to validate the atmospheric model used to drive FLEXPART-WRF (Advanced Regional Prediction System (ARPS); [10,11]). Furthermore, the occurrence of a complete daytime planetary boundary layer (PBL) cycle during the Saul's Creek fire (PBL mode transitioning from stable to convective during the morning and back to stable in the evening), as opposed to the consistently well-mixed Lehigh Gap case, allows for evaluation of VI diagnostic potential across a wider range of VI values than in [8].

Model simulations of the background meteorology and smoke plume behavior on 10 September 2018, the day of maximum prescribed fire activity, were utilized in this study to address two research questions. First, is a horizontal grid spacing of 4 km comparable to the finest grid spacing currently used in operational models (e.g., the 4-km North American Mesoscale (NAM) [12,13] and 3-km High Resolution Rapid Refresh (HRRR) [14,15]), sufficient to capture the spatiotemporal variability in wind and PBL structure during the Saul's Creek fire, or is finer grid spacing necessary? This question has relevance to prescribed burning in the western US, as models with O(4 km) grid spacing are critical components of NWS fire weather forecasts of VI and other metrics used by land managers to assess smoke dispersion potential [16]. Second, what is the relationship between VI and smoke dispersion during the Saul's Creek fire event, as measured by particle RT within a given horizontal or vertical distance from each particle's release point? Furthermore, are the current Colorado VAR categories an adequate indication of dispersion potential, as measured by RT? Given the aforementioned limitations of [8], application of the VI evaluation framework developed for eastern Pennsylvania to southwestern Colorado constitutes a critical step in the overall assessment of VI diagnostic potential.

It is important to note that we are not simulating the prescribed fire event in this study, but are instead performing a VI assessment on a day and in a location where a prescribed fire was conducted. The FLEXPART-WRF particle release strategy is intended to illustrate the differences in dispersion between different areas of complex terrain, not to reproduce the observed smoke plume. As in [8], the decision to use the setting and date of a prescribed fire for this study is based in part on the desire to examine smoke dispersion in complex terrain during actual prescribed fire weather conditions, and to take advantage of the available meteorological measurements. In this study, ARPS was used to simulate the background (i.e., no-fire) wind and PBL structure during the event. The ARPS simulations were validated using a suite of meteorological measurements, and ARPS output is used to drive FLEXPART-WRF. Finally, particle dispersion simulated by FLEXPART WRF was compared to VI computed from the ARPS output.

The remainder of this manuscript is organized as follows. The study methods are presented in Section 2, including a description of the southwestern Colorado study area and a brief summary of the prescribed fire event (Section 2.1), a summary of the numerical models and experiment methodology (Sections 2.2 and 2.3), and a description of the analysis methodology (Section 2.4). Results and discussion of the experiments are presented in Section 3, beginning with validation of the ARPS model (Section 3.1), and proceeding to an analysis of simulated thunderstorm outflows during the event (Section 3.2), an overview of particle behavior in the FLEXPART-WRF simulations (Section 3.3), and an analysis of the relationship between RT and VI (Section 3.4). Finally, the paper is concluded in Section 4.

## 2. Materials and Methods

*2.1. Location Description and Fire Event Summary*

As stated in Section 1, the focus of this study is smoke dispersion from a prescribed fire in the Saul's Creek area of the San Juan National Forest in southwestern Colorado. The study area is depicted in Figure 1 at three zoom levels, corresponding to the three ARPS model domains discussed in Section 2.2. The San Juan National Forest consists of approximately 1.8 million acres of federal lands in southwestern Colorado, with terrain varying from high-desert mesas to alpine peaks. The Saul's Creek area of San Juan National Forest, approximately co-located with the green square in Figure 1a,b, is situated immediately southwest of an area of 3000–4000 m surface elevations constituting the southwestern limit of the Rocky Mountains in Colorado. As seen at the innermost zoom level in Figure 1c, the terrain in the study area slopes broadly upward from southwest to northeast, with river valleys and isolated ridgelines complicating the local terrain pattern.

The Saul's Creek prescribed fire was primarily conducted within two burn units of the Columbine Ranger District (units 10 and 11; see black polygons in Figure 1c). The primary goal of the Saul's Creek prescribed fire was to reduce overall wildland fire hazards and reduce the likelihood of stand-replacing crown fires (Chris D. Tipton, personal communication). Secondary goals included increasing the resiliency of Ponderosa pine forests and increasing the probability of natural Ponderosa pine regeneration. Operations began on 8 September 2018 and ended on 12 September 2018, with blacklining occurring on 8–9 September 2018, aerial ignitions occurring on 10 September 2018, and holding and monitoring occurring on 11–12 September 2018; the peak day for burning was 10 September 2018 when approximately 400 ha were burned (over 50% of the total area burned). Estimated fire spread rates on 10 September 2018 were between 0.25 and 0.5 m min$^{-1}$, except during the afternoon when thunderstorm outflows increased spread rates to about 1 m min$^{-1}$. Between 12:00 mountain daylight time (MDT = UTC − 6) and 14:00 MDT, showers and thunderstorms developed north and northwest of the Saul's Creek site, propagating toward the southeast (not shown). The showers and thunderstorms largely dissipated before reaching the burn units, but thunderstorm outflows moved through the area along with light precipitation around 14:30 MDT.

Meteorological instrumentation was deployed during the Saul's Creek fire event with the specific goal of measuring the variables required for calculation of VI. A portable rawinsonde system was deployed immediately northeast of burn units 10 and 11 (triangle in Figure 1c) to collect profiles of potential temperature, mixing ratio, and wind speed at multiple times during the fire event day (09:00, 10:00, 12:00, and 14:00 MDT on 10 September 2018). A Vaisala CL31 ceilometer (Vaisala Corporation, Helsinki, Finland) was deployed coincident with the rawinsonde launch site to estimate mixing heights approximately every 15 s. The ceilometer is a light detection and ranging (LiDAR) instrument that measures backscatter, with three mixed layer height estimates derived from backscatter returns [17–19]. In the post-processing of the ceilometer backscatter data, an algorithm is used that searches for the three strongest negative backscatter gradients (i.e., backscatter sharply decreasing with increasing height). These gradients separate layers with relatively uniform aerosol density. Under clear-sky conditions, the tops of these aerosol layers can be interpreted as mixed layer height estimates. In addition to the instrumentation deployed during the fire event for VI calculation, Remote Automated Weather Stations (RAWS) surface observations (squares in Figure 1c) and Grand Junction, Colorado rawinsonde soundings (circle in Figure 1a) provide additional weather data with which to characterize the background meteorology. All meteorological observations are utilized in this study to evaluate the ability of the ARPS simulations to characterize the background weather conditions during the event (Section 3.1).

Two smoke monitors were set up north and west of the burn units to collect smoke concentration data. Due to the particle release strategy implemented in this study, in which the number of particles and release area are chosen with statistical evaluation of VI in mind, direct comparison of particle concentration between FLEXPART-WRF simulations and smoke concentration measurements is not

possible. Furthermore, the smoke monitors were largely unaffected by smoke on 10 September 2018, due to the prevailing southwest wind direction on that day (not shown).

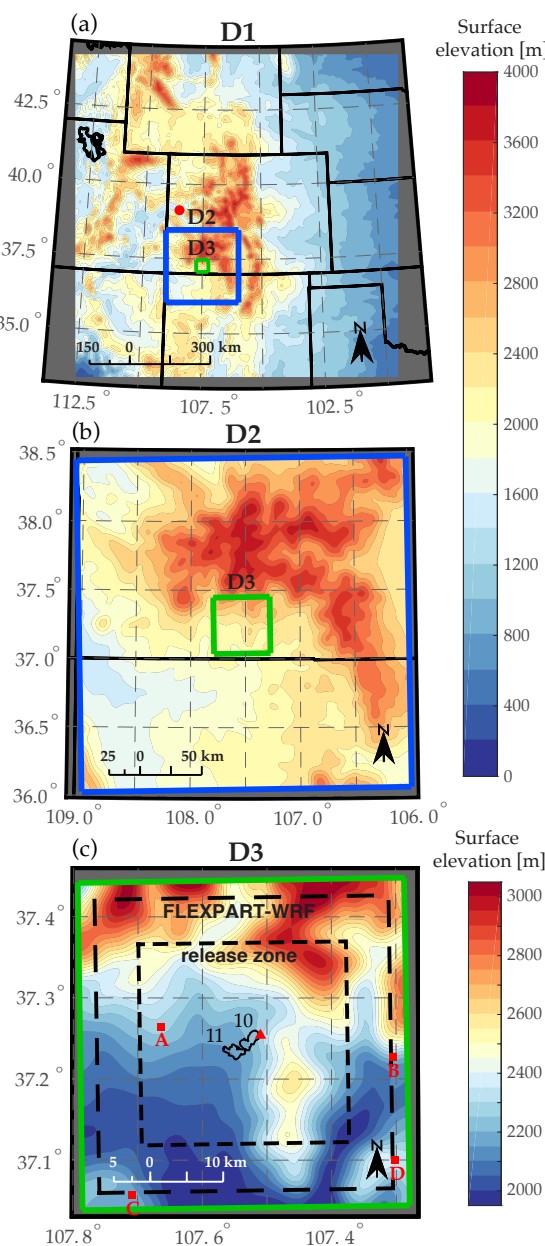

**Figure 1.** Plan view maps of surface elevation (m, above mean sea level) in Advanced Regional Prediction System (ARPS) domains (**a**) D1, (**b**) D2, and (**c**) D3. In (**c**), the FLEXPART-WRF domain and particle release zone described in Section 2.3 are indicated with long and short dash black squares, respectively. In (**a**,**c**), red symbols indicate observation sites: circle Grand Junction, Colorado (KGJT) rawinsonde, triangle (Saul's Creek portable rawinsonde and light detection and ranging (LiDAR) ceilometer, co-located), square (Remote Automated Weather Stations (RAWS); letters correspond to panels in Figure 5). The black polygons in the center of (**c**) depict the two burn units active on 10 September 2018, units 10 and 11.

## 2.2. ARPS Model Configuration and Parameterization

As this study builds upon the previous FLEXPART-WRF VI assessment study [8], the methodology detailed in this and the following section is similar to that presented therein. Data from this study are freely available for download at http://eams2.usfs.msu.edu/Pub_Data/outgoing/kiefer/covi/.

In this study, FLEXPART-WRF was driven by simulated meteorological fields from ARPS version 5.3.4. ARPS has been shown in previous studies to be applicable to atmospheric phenomena ranging in scale from meso- to synoptic-scale ([11,20–22], for example) to micro-scale [23–25]. As ARPS is a three-dimensional, compressible, non-hydrostatic atmospheric model with a terrain-following coordinate system, it has been applied to flows in complex terrain in a number of previous studies, including [8,21,26,27]. The choice to use ARPS in this study was based primarily on the established utility of ARPS for simulating flows in complex terrain as well as the implementation of terrain shadowing effects in ARPS by [28].

In this study, ARPS was executed in one-way nested mode with three domains, beginning with an outermost domain over the western US, with 4000-m horizontal grid spacing ("D1"), and nesting down through an intermediate 1333-m horizontal grid spacing domain ("D2"), to the innermost domain, with 444-m horizontal grid spacing ("D3") (Figure 1 and Table 1). A stretched vertical grid was used in all three domains, with minimum vertical grid spacing at the ground of 50 m, 25 m, and 10 m in domains D1, D2, and D3, respectively. Initial and boundary conditions for the outermost domain were provided by the 12-km NAM model [12]. For the outer two domains, US Geological Survey (USGS) terrain data (30-arc s) (url: http://www.caps.ou.edu/ARPS/arpsdown.html) interpolated to the model grid are utilized, whereas for the innermost domain, Shuttle Radar Topography Mission (SRTM) terrain data (3-arc s) (url: https://dds.cr.usgs.gov/srtm/version2_1/SRTM3/) interpolated to the model grid are implemented. SRTM terrain data are used in place of the USGS data as the latter data source was found by [8] to misrepresent some topographic features at spatial scales on the order of 500 m or less, topographic features of potential relevance to local smoke dispersion. For all domains, the USGS land cover data bundled with ARPS were utilized; however, the circa-1992 USGS leaf area index (LAI) and vegetative fraction data were replaced with Moderate Resolution Spectroradiometer (MODIS) collection 6 LAI and Fraction of Photosynthetically-Active Radiation absorbed by vegetation (FPAR) data, respectively [29]. The 500-m pixel size LAI and FPAR estimates were derived from data collected by the National Aeronautics and Space Administration (NASA) Terra and Aqua satellites during the eight-day period from 8–15 September 2018.

**Table 1.** ARPS nested grid configurations (domains D1, D2, and D3) and FLEXPART-WRF grid configuration (domain FW), with dimensions (excluding boundary points). $\Delta x$ and $\Delta y$ are the horizontal grid spacing in the west–east and south–north directions, $\Delta z_{min}$ is the vertical grid spacing of the first grid cell above the ground, and $N_{500}$ and $N_{3000}$ are the number of vertical grid levels within the lowest 500 m and 3000 m of the model atmosphere, respectively (values chosen based on evaluation of simulated mixed layer heights during morning and afternoon). For reference, the lowest model level for scalars and horizontal wind components is one-half of $\Delta z_{min}$.

| Domain | Grid Size | Domain [km] | $\Delta x$, $\Delta y$ [m] | $\Delta z_{min}$ [m] | $N_{500}$ | $N_{3000}$ |
|--------|-----------|-------------|-----------|--------------|-----------|------------|
| D1 | $300 \times 300 \times 50$ | $1200.0 \times 1200.0 \times 20.0$ | 4000.0, 4000.0 | 50.0 | 10 | 25 |
| D2 | $200 \times 200 \times 50$ | $266.6 \times 266.6 \times 20.0$ | 1333.0, 1333.0 | 25.0 | 12 | 26 |
| D3 | $100 \times 100 \times 50$ | $44.4 \times 44.4 \times 20.0$ | 444.0, 444.0 | 10.0 | 14 | 26 |
| FW | $90 \times 90 \times 50$ | $40.0 \times 40.0 \times 18.9$ | 444.0, 444.0 | 10.0 | 13 | 25 |

In all domains, a 1.5-order subgrid-scale turbulence closure scheme with a prognostic equation for the turbulent kinetic energy (TKE) was utilized [30], as well as a land surface and vegetation model based on [31,32] and radiation physics following [33–35]; in all domains, a non-local turbulence parameterization based on [36] was also applied. Effects of topographic shading on radiative fluxes were accounted for following [28]. All three domains were initialized at 18:00 MDT 09 September 2018 and run for 30 h; simulated variables were output from the D3 simulation at 1-min frequency and were subsequently used to drive FLEXPART-WRF; however, the first 14 h of the D3 simulation were not utilized by FLEXPART-WRF, as they served the sole purpose of allowing a nocturnal PBL to develop and evolve during the night prior to the 10 September 2018 event.

### 2.3. FLEXPART-WRF Model Configuration, Parameterization, and Experiment Design

FLEXPART-WRF is a version of the Lagrangian particle dispersion model FLEXPART [37], originally developed for use with global weather models, that has been adapted for use with limited area models, specifically the Weather Research and Forecast (WRF) model [38]; this study used FLEXPART-WRF version 3.3. We wish to make it clear that although we refer to the model by its proper name "FLEXPART-WRF", the model in this study is actually driven by output from the ARPS model. In addition to the previous ARPS–FLEXPART-WRF study in eastern Pennsylvania [8], FLEXPART-WRF has also been coupled to ARPS-CANOPY, a version of ARPS with a canopy parameterization, and used to simulate smoke dispersion during a low-intensity prescribed fire field experiment [39]. In the latter study, the authors concluded from a comparison of smoke plume simulations and field observations that the model was capable of reproducing some of the observed variations in smoke concentrations and plume heights. The choice to use FLEXPART-WRF in this study was based mainly on its Lagrangian reference frame, which [40] argue is more suitable for use in complex terrain than Eulerian dispersion models since Lagrangian dispersion models can dynamically compute plume rise and are more computationally efficient at small spatial scales. Also, FLEXPART-WRF has been applied previously to fire and smoke dispersion studies (e.g., [39]), including those in complex terrain (e.g., [41]).

FLEXPART-WRF uses a Lagrangian reference frame, tracking individual particles and updating particle position at every time step based on a mean wind component and a random wind component. The mean and random wind components are associated with mean transport and turbulent diffusion, respectively. The mean wind is computed in FLEXPART-WRF from three-dimensional wind components ingested from the meteorological model, with the user given the choice of using instantaneous winds, time-averaged winds, or instantaneous horizontal wind components with a diagnostic calculation of vertical velocity. Since ARPS does not output time-averaged winds (as in WRF versions 3.3 or higher), the instantaneous wind option was utilized in this study. The random wind component was computed either via the Hanna turbulence parameterization [42], or computed as a function of TKE read in from the meteorological model. The Hanna parameterization used in FLEXPART-WRF is identical to the version used in the original FLEXPART model, and is strongly recommended by [1] based on their assessment that the ingested TKE option violates the well-mixed criterion. Thus, we followed best practices and used the Hanna turbulence formulation, which is a function of three PBL parameters: mixed layer height, friction velocity, and surface heat flux. In order to assure consistency between the mixing height calculations used in ARPS and FLEXPART-WRF, we used the FLEXPART-WRF option directing the model to use PBL parameters ingested from ARPS, rather than the option directing FLEXPART-WRF to compute the PBL parameters internally.

The FLEXPART-WRF simulation was initialized at 08:00 MDT 10 September (14 h after ARPS initialization), and ran for a total of 11 h, ending at 1900 MDT. The start and end time for the FLEXPART-WRF simulation were chosen to overlap with the period of hourly ground-based smoke plume observations. The FLEXPART-WRF grid was consistent with the ARPS D3 (i.e., innermost) grid, with the same horizontal and vertical grid spacing. However, the FLEXPART-WRF grid extends five grid points fewer in each horizontal direction to avoid driving the particle model with output from the ARPS lateral boundary relaxation zone (Figure 1 and Table 1). ARPS variables were read once every minute from NetCDF files that were generated from ARPS2FLEX, a routine that transforms the output from ARPS into the suite of variables that FLEXPART-WRF is designed to read in from WRF output. Three-dimensional particle position was updated every 10 s, and written out to ASCII files at the same interval; particle concentration was also computed but is not presented here. A total of 275,000 particles were released at a constant rate during the 11-h FLEXPART-WRF simulation, across an approximately 28-km × 28-km zone centered on the two burn units active on 10 September 2018 (Figure 1c), within the lowest grid layer. The particle release zone represents a compromise between the desire to maximize terrain heterogeneity across the release zone and the self-imposed requirement that there be a minimum distance of 5 km between the outermost particles in the release zone and the domain lateral boundaries. Initial particle position within the release zone was specified randomly.

Particle diameter was restricted to the range 2.5 μm or smaller (i.e., PM$_{2.5}$), with a mean particle diameter of 0.5 μm, and a particle density of 1.358 g cm$^{-3}$ ([43], representative of a mixture of longleaf needle and Juniper/rabbitbrush/sage). Deposition processes were engaged in FLEXPART-WRF; however, in this study, only dry deposition was applied. It is worth noting that the release strategy used in this study is designed to allow a comparison of dispersion characteristics at different locations during the time the Saul's Creek fire occurred, and is not intended to represent actual emissions during the burn.

*2.4. Analysis Methodology*

As a preliminary step in the analysis of particle dispersion, RT was computed for each of the 275,000 particles released during the FLEXPART-WRF simulation. For each particle, horizontal RT (HRT) was computed within a 5-km radius circle centered on the particle release point, and vertical RT (VRT) was computed within a 250-m deep layer above the particle release level. The decision to use 5 km and 250 m for horizontal radius and depth, as opposed to 1 km and 100 m as in [8], was based on both the broader scale of topography in southwestern Colorado compared to eastern Pennsylvania and the use of coarser grid spacing in the current study. Upon reaching the given radii/depths, particles crossed approximately the same number of FLEXPART-WRF grid cells in both studies. The relationship between VI and HRT (VRT) was examined for alternate radii (depths) of 1 km and 10 km (100 and 500 m), with limited sensitivity of results to choice of radii (depth) (not shown). As in [8], longer (shorter) RT is equated with weaker (stronger) particle dispersion, and the terms are used interchangeably. Note that although VI does not indicate the preferred direction of dispersion (horizontal vs. vertical), the relationship of VI to horizontal and vertical dispersion was considered separately, in order to examine if VI is more closely related to one or the other.

VI was computed as the product of a mixing height (MH), defined as in [8] as the height above the surface where virtual potential temperature first exceeds the lowest grid level value, and a transport wind speed, defined here as the numerical average wind speed within the mixed layer (WS$_{ML}$). Also as in [8], a 1 K perturbation was added to the lowest grid level virtual potential temperature to ensure that the bases of weak inversions are not misidentified as the mixing height. Note that the virtual potential temperature was otherwise identical to potential temperature, except that temperature was replaced by virtual temperature, the temperature that dry air would have if its pressure and density were equal to those of a given sample of moist (unsaturated) air (url: http://glossary.ametsoc.org/wiki/Virtual_potential_temperature). As pointed out in [8], the methods chosen for calculating MH and WS$_{ML}$ are consistent with current NWS forecast practices [44]; however, the 1 K increment used in both studies is larger than the increment that is currently implemented at the NWS (0.5 K) [44], but was determined to be the smallest value that consistently bypassed weak inversions in the model [45], yielding spatially consistent MH. When interpreting mixing height computed in this study, the reader should keep in mind two points. First, mixing height may be overestimated to some degree by the use of a 1 K perturbation to enhance parcel buoyancy, rather than the 0.5 K perturbation used by the NWS. Second, the calculation of mixing height using the lowest model level (D1: 25 m above ground level (AGL); D2: 12.5 m AGL; D3: 5 m AGL) virtual potential temperature may lead to an underestimation of surface buoyancy and an underestimation of mixing heights, since shallow superadiabatic layers at the surface are bypassed. Note that an alternative mixing height calculation based on TKE, identified by [45] as the most accurate of four methods considered in their study, has been tested in this study (not shown); however, due to uncertainty in interpreting mixing heights from ARPS parameterized TKE, further examination is left to future work.

## 3. Results and Discussion

### 3.1. Meteorological Model Validation

Before proceeding to the assessment of FLEXPART-WRF simulated particle dispersion and VI diagnostic value, it is important to establish whether the ARPS simulations capture the spatiotemporal variations in meteorological variables necessary for FLEXPART-WRF to simulate plume behavior with fidelity. This exercise proceeds in a telescoping manner. First, the vertical structure of the troposphere and lower stratosphere at the nearest upper-air network site approximately 225 km northwest of the Saul's Creek fire (red circle in Figure 1a) is examined. Second, the vertical structure of the mixed layer at a portable rawinsonde site approximately 250 m northeast of burn unit 10 (red triangle in Figure 1c) is evaluated. Third, the temporal evolution of mixed layer height based on portable rawinsonde soundings and LiDAR ceilometer backscatter data collected at the portable rawinsonde site is assessed. Fourth, the temporal evolution of surface meteorological variables at four RAWS stations within 10–25 km of the Saul's Creek fire (red squares in Figure 1c) is examined.

First, vertical soundings of temperature and dewpoint are presented on a skew T–log P diagram, along with hodographs and wind vectors, as observed at Grand Junction, Colorado (KGJT, see red circle in Figure 1a), and as simulated by ARPS in domain D1, at 18:00 MDT 9 September, 06:00 MDT 10 September, and 18:00 MDT 10 September (Figure 2). As shown in Figure 2, the observed and model soundings generally agree, with the model simulating a deep mixed layer at the 18:00 MDT observation times (cf. Figure 2a–c,f) and a stably-stratified nocturnal boundary layer beneath a residual layer at the 06:00 MDT 10 September observation time (Figure 2b,e). Furthermore, the model captures the west to northwest winds in the mixed layer at the 18:00 MDT observation times, and east to southeast winds in the nocturnal boundary layer at the 06:00 MDT 10 September observation time. With respect to simulation deficiencies, a consistent cold bias at the surface is seen at the 18:00 MDT observation times, and the model fails to reproduce a saturated layer atop the mixed layer (with rapid changes in dewpoint in the overlying free atmosphere, evidence of cumulus cloud development atop the mixed layer) at 18:00 MDT 10 September.

Proceeding to vertical profiles at the portable rawinsonde site northeast of burn units 10 and 11 (Figure 3), the vertical structure of virtual potential temperature ($\theta_v$) and wind speed and direction ($S$, $V_H$) are generally captured by all three ARPS domains. First, the $\theta_v$ profile is largely insensitive to changes in horizontal grid spacing and minimum vertical grid spacing near the surface (Table 1), although the superadiabatic layer evident at 14:00 MDT (Figure 3d) becomes progressively stronger with decreasing grid spacing. Although the ARPS simulations capture the evolution of the $\theta_v$ profile from a morning configuration (Figure 3a) to a daytime configuration (Figure 3d), it is worth noting the persistent 3–5 °C cold bias near the surface, and the absence during the morning hours of an inversion with a base just below 4 km above ground level (AGL) (Figure 3a,b). The latter feature appears to be a capping inversion above a residual layer remaining from the previous day's mixed layer.

Regarding the evolution of the vertical wind structure, all three ARPS domains capture the evolution from a two-layer profile at 09:00 MDT (Figure 3a), with weak wind speeds ($<2$ m s$^{-1}$) and variable wind direction below approximately 1.5 km AGL, and uniform northwest winds above 1.5 km AGL, to a profile at 14:00 MDT with winds smoothly veering with height from southwest to northwest (Figure 3d). The evolution in vertical wind structure in the lowest 1.5 km reflects the evolution from a nocturnal boundary layer in complex terrain, in which intermittent turbulence and multi-scale topographic features yield winds highly variable in time and space, to a daytime convective boundary layer, in which daytime mixing and the broadly sloping terrain (Figure 1c) yields southwest flow less variable in time and space than in the nocturnal boundary layer. It is worth noting that the ability of the model to capture the vertical variation in wind direction during the morning hours is sensitive to grid spacing, with domains D2 and D3 comparing better with the observed profile than domain D1 (Figure 3a,b). However, during the afternoon hours, differences between the three model domains are more subtle and all three domains capture the observed vertical structure similarly (Figure 3c,d).

Finally, the absence in the model profiles of a weak low-level jet between approximately 2–4 km AGL, observed at 09:00, 10:00, and 12:00 MDT (Figure 3a–c), may be related to the absence in the model simulations of the aforementioned capping inversion atop the residual layer.

Next, time series of simulated mixed layer height are compared to mixed layer height time series derived from the portable rawinsonde soundings (Figure 3) and LiDAR ceilometer backscatter data (Figure 4). Simulated and observed mixed layer height depict a broadly similar evolution prior to thunderstorm outflow passage, with mixed layer height increasing from less than 1 km before 12:00 MDT to 2–4 km by approximately 15:00 MDT. Note that the ceilometer detects mixed layer heights in the 2–4 km range between 10:00 and 12:00 MDT, likely evidence of a residual layer remaining from the previous day's daytime PBL. There is a considerable disparity in observed mixed layer heights at the 12:00 and 14:00 rawinsonde launch times between values derived from the portable rawinsonde soundings and values derived from the ceilometer backscatter returns. Given that ceilometer-derived mixed layer heights reach 3–4 km by 17:00–18:00 MDT, and residual layer heights associated with the previous day's PBL are in the same range, it appears that the parcel method used to derive mixed layer height from virtual potential temperature overestimates the increase in mixed layer height between 10:00 and 12:00 MDT. The rapid decrease in ceilometer-derived mixed layer height between 15:00 and 15:30 MDT, and model-simulated mixed layer height between 17:00 and 17:30 MDT (domains D2 and D3 only) is evidence of the passage of thunderstorm outflows (the final rawinsonde launch was at 14:00 MDT, prior to outflow passage). Finally, note that peak mixed layer heights are approximately 1 km greater in domains D2 and D3 than in domain D1, likely the result of partially-resolved PBL processes in the inner domains.

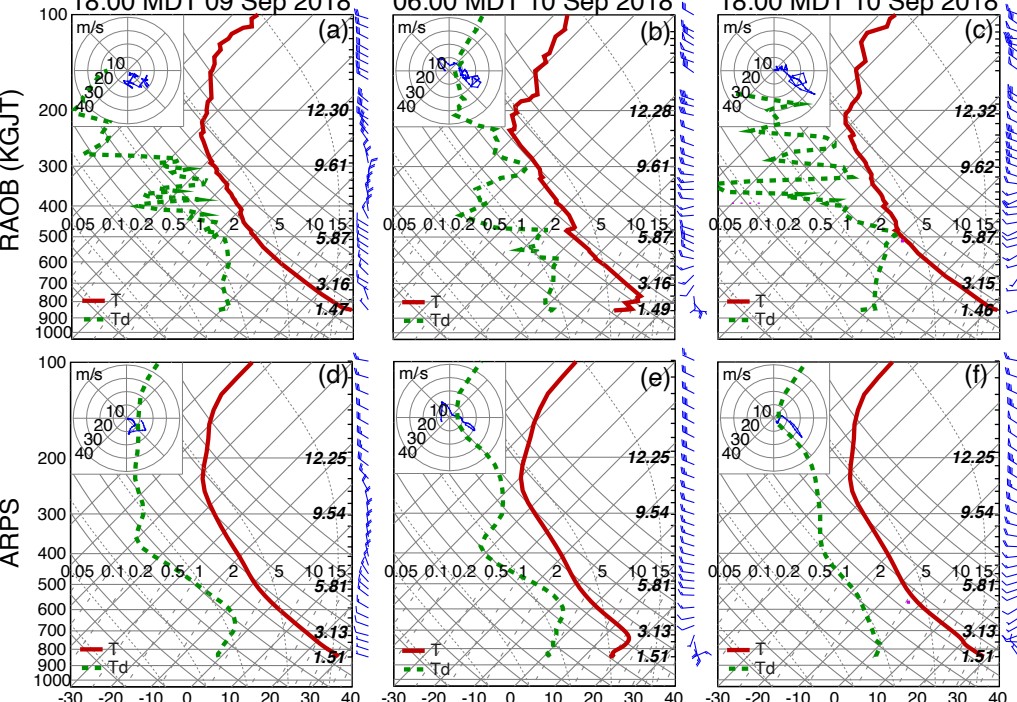

**Figure 2.** Skew T–log P comparison of (**a**–**c**) KGJT rawinsonde sounding, and (**d**–**f**) D1 model sounding at the nearest grid point to KGJT. In all panels, the red line is temperature and the green line is dewpoint; the blue curve in the top-left corner of each panel indicates the hodograph trace. The values on the x-axis and y-axis denote temperature (°C) and atmospheric pressure (hPa), respectively; approximate height above mean sea level (km) is indicated for select pressure levels on right side of each panel. Location of KGJT is indicated by a red circle in Figure 1a.

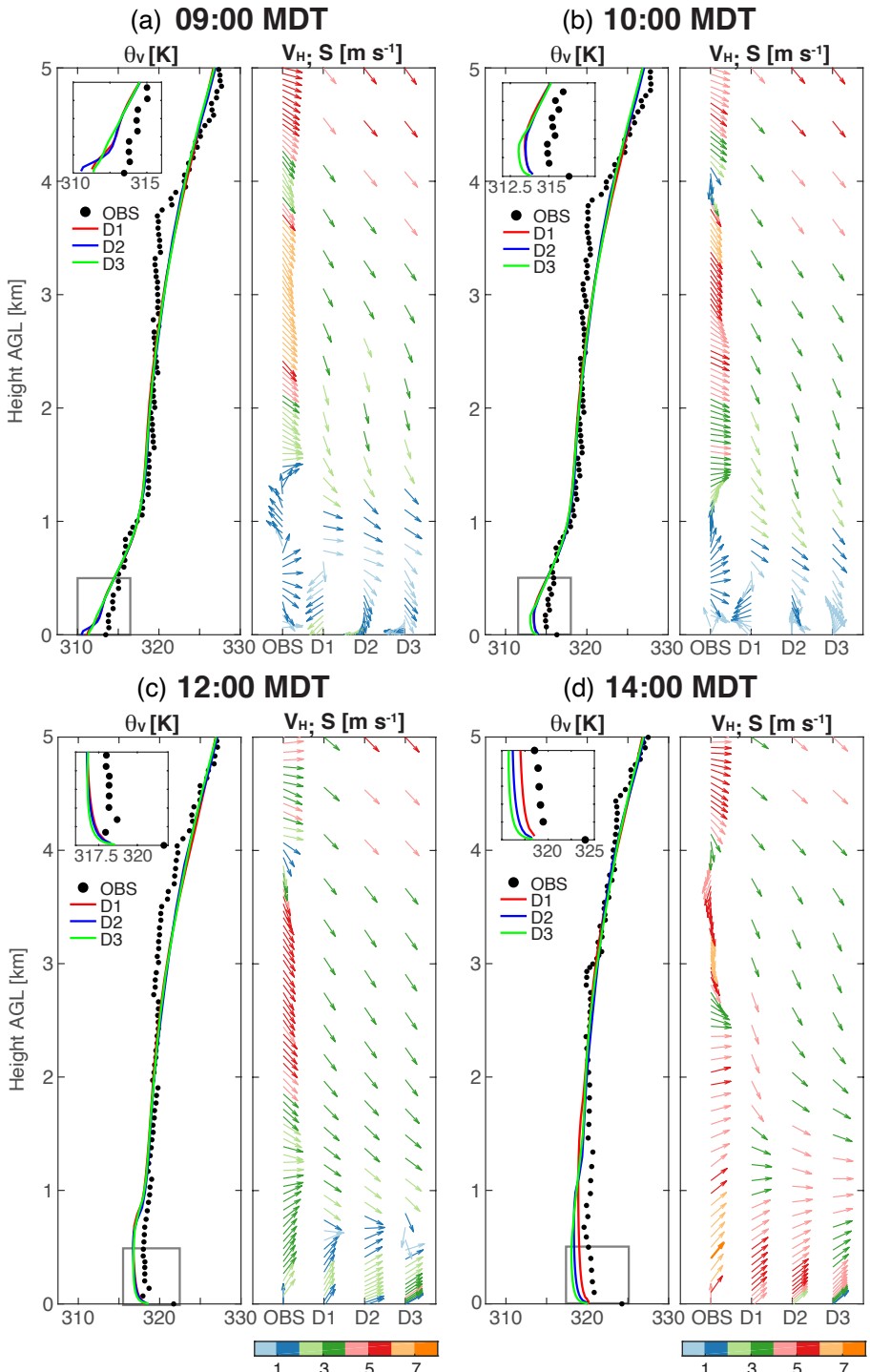

**Figure 3.** Vertical profile comparison of portable rawinsonde sounding and model sounding at nearest grid point to sounding launch site (red triangle in Figure 1c), at (**a**) 09:00, (**b**) 10:00, (**c**) 12:00, and (**d**) 14:00 MDT 10 September 2018, for all three model domains. In each panel, left sub-panel depicts virtual potential temperature, and right sub-panel depicts wind vectors (color scale for wind speed included at bottom of figure). In order to focus on the planetary boundary layer (PBL), only the lowest 5 km of the atmosphere is displayed. Inset panels in upper-left corner of each virtual potential temperature panel depict the lowest 500 m of the model atmosphere.

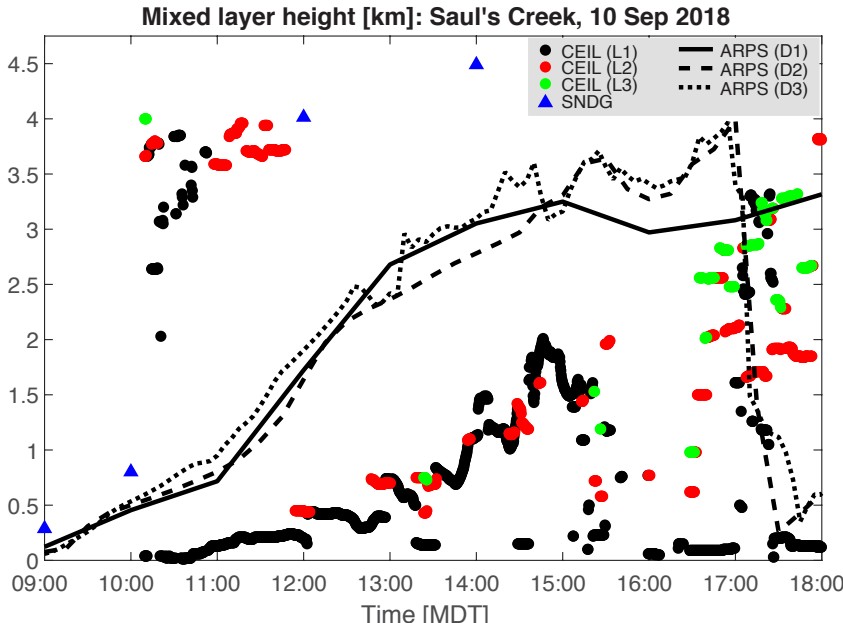

**Figure 4.** Time series comparison of observed mixed layer height from ceilometer aerosol backscatter returns (CEIL (three aerosol layers): black, red, and green circles) and portable rawinsonde soundings (SNDG: blue triangles), and model estimated mixed layer height from soundings extracted at nearest grid point to center of two burn units depicted in Figure 1c (ARPS (domains D1-D3): lines). Sounding-based mixed layer height is defined in this study as the height above the surface where virtual potential temperature first exceeds the lowest-level value, plus a 1-K perturbation; see text for details. See Section 2.1 for details on the relationship of ceilometer aerosol layers and mixed layer height estimates.

Finally, time series of observed and simulated surface temperature, dewpoint, and wind speed and direction are examined at four RAWS stations within 25 km of the Saul's Creek fire, during the 30-h period from 18:00 MDT 9 September–00:00 MDT 11 September (Figure 5). Note that due to differences in the height AGL of the observed and simulated variables, caution should be exercised when directly comparing values between observations and simulations. Overall, ARPS captures the diurnal temperature cycle, albeit with an underestimated diurnal range; differences between model domains are small outside of the period from 17:00–20:00 MDT 10 September (see discussion of thunderstorm outflows in Section 3.2). A consistent cold bias of 3–5 °C is seen during the day, with bias generally smaller during the night (approximate local sunrise/sunset: 06:48/19:26 MDT 10 September). No consistent dewpoint bias is noted in the model simulations, although the aforementioned model cold bias implies a positive relative humidity bias during the day (not shown). The evolution of wind during the 30-h period follows a similar pattern at all four RAWS sites, with east to northeast winds before approximately 12:00 MDT 10 September, west to southwest winds between 12:00 MDT and about 15:00–16:00 MDT, and north to northeast winds after 16:00 MDT. The model simulations capture this general evolution, with the notable exception of the period from about 15:00 MDT to about 18:00 MDT when model error in thunderstorm outflow timing yields large differences in wind speed and direction between model and observations. At the San Juan Portable RAWS site (Figure 5a, closest to burn units 10 and 11), two wind shifts and temperature drops are seen in the observed time series (14:00 and 20:00 MDT), whereas the model simulations depict a single wind shift and temperature drop (19:00 MDT in D1 and 17:00 MDT in D2 and D3).

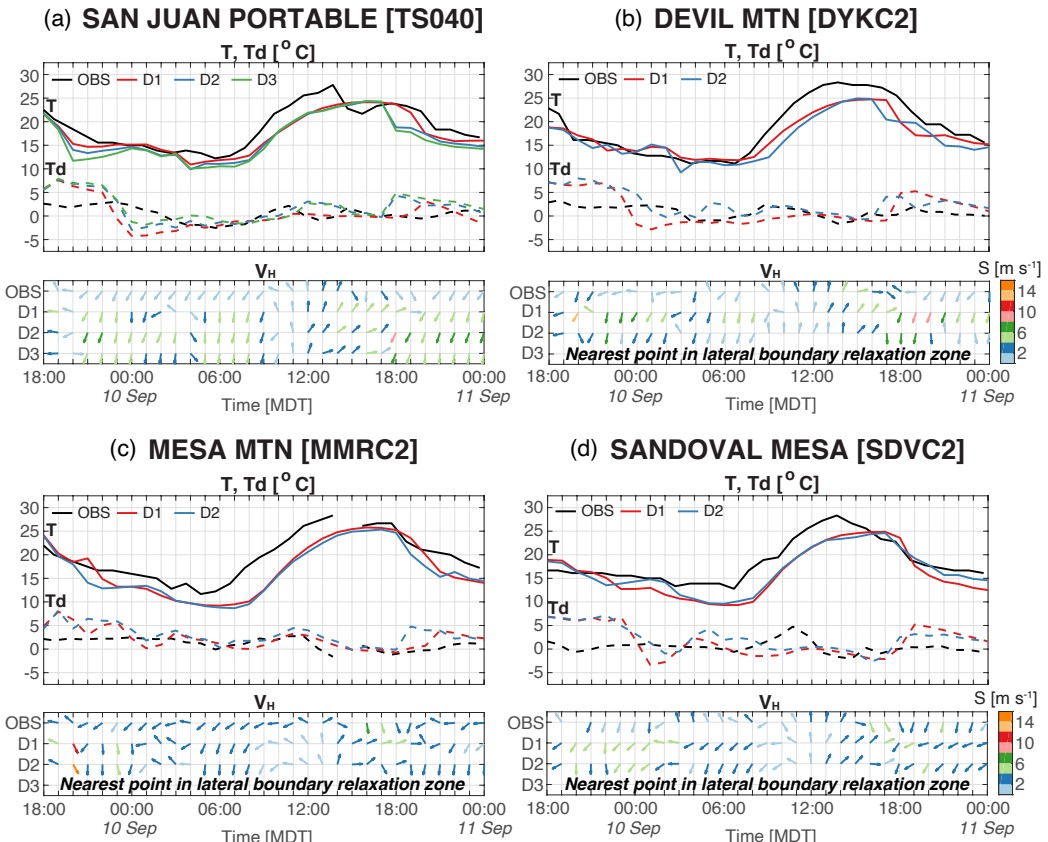

**Figure 5.** Time series of observed and simulated temperature (T), dewpoint (Td), and horizontal wind vectors ($V_H$; color scale for wind speed (S) included to right of figure), at four RAWS sites: (**a**) San Juan Portable (TS040), (**b**) Devil Mtn (DYKC2), (**c**) Mesa Mtn (MMRC2), and (**d**) Sandoval Mesa (SDVC2). Note that domain D3 is omitted from panels (**b**–**d**) because nearest grid points to RAWS sites are in lateral boundary relaxation zone. Temperature and dewpoint are measured at 2 m above ground level (AGL) at all four RAWS stations, and wind speed is measured at either 2 m AGL (TS040) or 6.1 m AGL (DYKC2, MMRC2, and SDVC2); all simulated variables are valid at the lowest grid level (D1: 25 m; D2: 12.5 m; D3: 5 m AGL). See Figure 1c for location of RAWS sites.

### 3.2. Thunderstorm Outflow Summary

Before proceeding with analysis of the FLEXPART-WRF simulation, it is prudent to examine the evolution of simulated near-surface temperature and wind prior to and during the passage of the aforementioned thunderstorm outflows. The purpose of this examination is two-fold: establish the timing and location of the simulated thunderstorm outflows during the afternoon, and assess the sensitivity of the timing and location to model grid spacing. As such, Figure 6 depicts lowest model level temperature and horizontal wind vectors between 15:00 MDT and 18:00 MDT. Beginning at 15:00 MDT (Figure 6a–c), the same spatial pattern is depicted in all three domains, consisting of southwest flow with temperature decreasing from southwest to northeast with increasing surface elevation (Figure 1c). The impact of the increasing resolution of topographic features like the south-north–oriented ridge southeast of burn units 10 and 11 is seen in the progressively increasing spatial variability of temperature and wind from left (D1) to right (D3).

From 16:00 MDT to 18:00 MDT, the passage of the thunderstorm outflow is depicted by the emergence of north to northeast winds and temperatures 5–10 °C lower than depicted in the pre-outflow environment at 15:00 EDT (cf. Figure 6a–l). As depicted in the simulated temperature time series at the San Juan Portable RAWS site (Figure 3a; location depicted with blue square in Figure 6), temperatures drop approximately 5 °C within one hour and winds shift from southwest to northeast and, in domains

D2 and D3, also increase in magnitude. As noted in the discussion of Figure 5 in Section 3.1, the timing of the thunderstorm outflow is sensitive to model grid spacing, with arrival of the temperature drop and wind shift differing by two hours between domain D1 (19:00 MDT) and domains D2 and D3 (17:00 MDT).

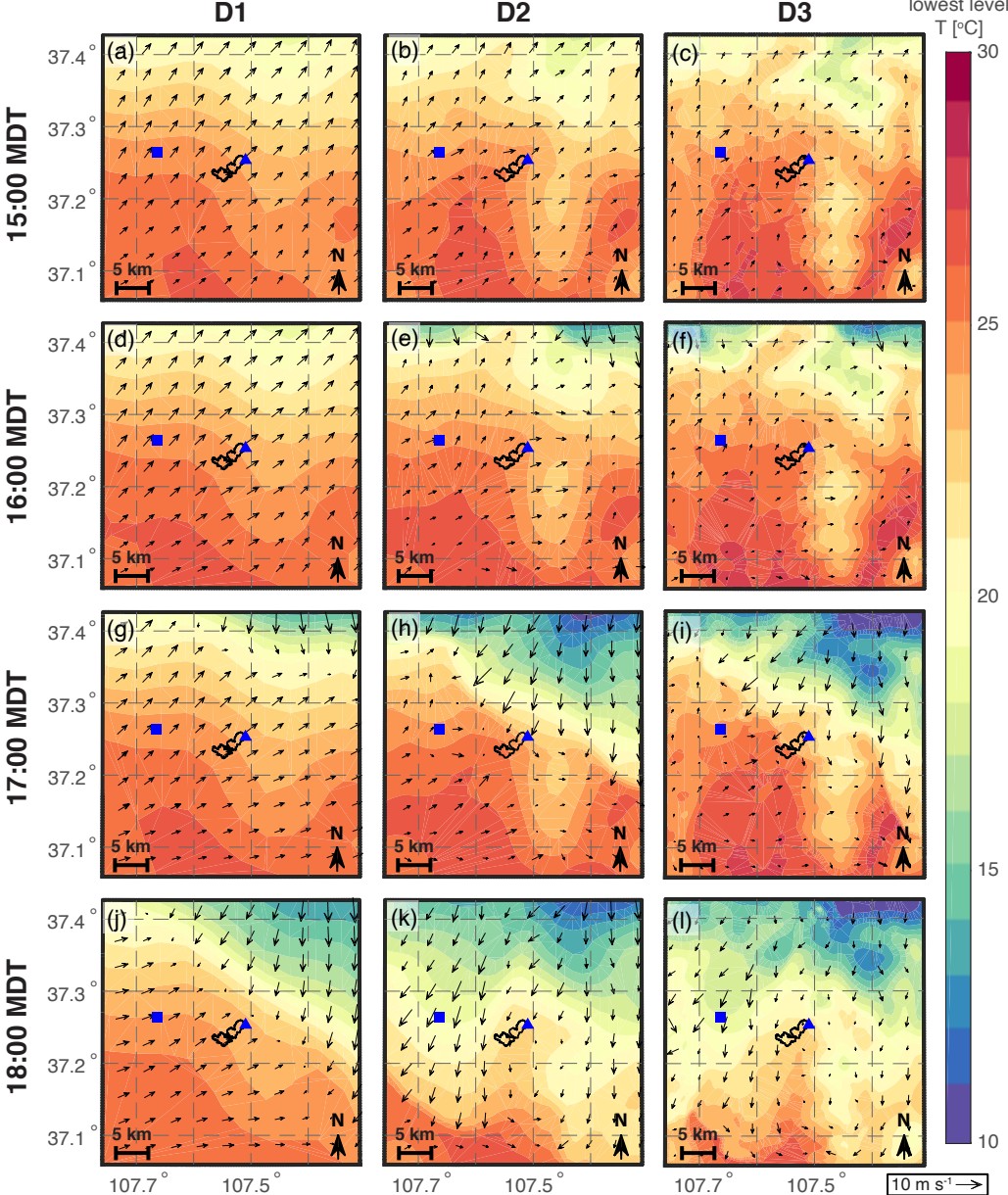

**Figure 6.** Contoured maps of lowest model level temperature and horizontal wind vectors between (**a**–**c**) 15:00 and (**j**–**l**) 18:00 MDT 10 September 2018, with a 1-h interval between rows, in domains (left) D1, (center) D2, and (right) D3; the same region (domain D3) is depicted for all three domains. Contour interval is 1 °C, and vector reference scale is included in lower-right corner of figure. The black polygons in the center of each panel depict the perimeters of the two burn units active on 10 September 2018, and the blue square and triangle indicate the locations of the San Juan Portable RAWS (Figure 5a), and portable rawinsonde and LiDAR ceilometer (Figures 3 and 4, respectively). For reference, the lowest model level in domains D1, D2, and D3 is 25 m, 12.5 m, and 5 m AGL, respectively (Table 1).

Finally, note the differences in temperature and wind spatial variability between model domains during and after thunderstorm outflow passage in Figure 6g–l. Such meteorological differences yield

differences in particle dispersion evolution in FLEXPART-WRF simulations driven by meteorological output from each ARPS domain simulation (not shown). The remainder of this manuscript focuses on the FLEXPART-WRF simulation driven by output from the domain D3 simulation. We choose to use the D3 output to drive FLEXPART-WRF for two reasons. First, the model assessment in Section 3.1 shows that the wind and PBL structure during the Saul's Creek event are best captured by the D3 simulation. Second, the expected evolution of numerical weather prediction toward sub-kilometer grid spacing forecasts motivates an assessment of VI computed from a sub-kilometer grid spacing model simulation. We leave examination of the sensitivity of VI diagnostic value to model grid spacing to future work.

### 3.3. Particle Behavior Overview

Figure 7 shows the initial particle position of all particles released during the simulation, with the HRT and VRT of each particle (Figure 7a,b, respectively) indicated by marker color. The particle plots in Figure 7 depict spatially heterogeneous patterns of HRT and VRT that appear to correspond to the pattern of surface elevation (dark gray contours in Figure 7). Despite sharing the common trait of a RT–topography correspondence, the patterns of HRT and VRT are not identical. Broadly speaking, HRT is shortest in the higher terrain across the northern third of the release zone (0–1 h), and longest along the west- and east-facing slopes of the prominent ridgeline southeast of burn units 10 and 11 (2+ h) (Figure 7a). In the southwest corner and atop the ridgeline, HRT is generally between 1–2 h. Regarding VRT (Figure 7b), RT is shortest in the western portion of the release zone (0–0.5 h), and longest along an axis from the southwest corner of the release zone to north of burn units 10 and 11 (1.5–2.5 h), with VRT on the order of 0.5–1.5 h along the north-south ridgeline and in the higher terrain across the north.

Following the strategy of [8], analysis zones are defined for the purpose of assessing VI diagnostic value in areas of the domain with relatively good ventilation and areas with relatively poor ventilation, and in areas of the domain with differing terrain characteristics. The following 5-km × 5-km analysis zones are defined (Figure 7), grouped by terrain characteristics (Mountain, Slope, Plains): $Z_{M1}$, $Z_{M2}$, $Z_{S1}$, $Z_{S2}$, $Z_{S3}$, and $Z_{P1}$. Although the other five zones were chosen on the basis of RT and topography, $Z_{S2}$ was chosen to represent the Saul's Creek burn area, and is centered on burn units 10 and 11. The reader is reminded that the fire was not simulated in this study; $Z_{S2}$ was intended to represent dispersion potential from the Saul's Creek site on the burn day.

With the analysis zones defined, a comparison of $PM_{2.5}$ "plumes" from each zone is presented in Figure 8. The term "plume" is used with caution here, as the "plume" for each analysis zone is visualized by isolating all particles released from that zone; recall that particles are released throughout the 28-km × 28-km release zone depicted in Figures 1c and 7. Considering all times and zones, a general diurnal pattern is evident. At 09:00:10 MDT (Figure 8, left column), north to northeast winds and stable stratification (Figure 3a) yield near-surface transport of particles toward lower elevations, with direction of transport varying between analysis zones and with little to no vertical movement of particles. At 13:00:10 MDT (Figure 8, center column), southwest to west winds through a deep mixed layer (Figures 3c,d and 4) yields lofting of particles and plumes directed toward the northeast and east. Finally, at 17:00:10 MDT (Figure 8, right column), the preceding hours of strong particle dispersion yield less concentrated plumes, again directed toward the northeast and east. The exception to this general behavior at 17:00:10 MDT is the behavior of particles released within $Z_{M1}$: north winds and stable stratification within the thunderstorm outflow (Figure 6i) yields a plume directed toward the south, with relatively little vertical mixing. One hour later, the outflow has passed through the remainder of the domain (Figure 6l), impacting dispersion in all analysis zones (not shown). Finally, it is worth noting that particle dispersion differences among analysis zones are greatest at 09:00:10 MDT, smallest at 13:00:10 MDT, and larger again at 17:00:10 MDT as thunderstorm outflows impact the northernmost zone ($Z_{M1}$) first.

It is worth noting that the evolution of the simulated smoke plume from $Z_{S2}$ (centered on the Saul's Creek site) agrees broadly with the smoke observations made during the event by a human observer. At 09:00 MDT, the observed smoke plume was laid down on the ground with little or no wind and a near-surface temperature inversion. At 13:00 MDT, the observed smoke plume was upright and directed toward the east. Finally, at 17:00 MDT, the observed smoke plume had thinned and was directed toward the southeast.

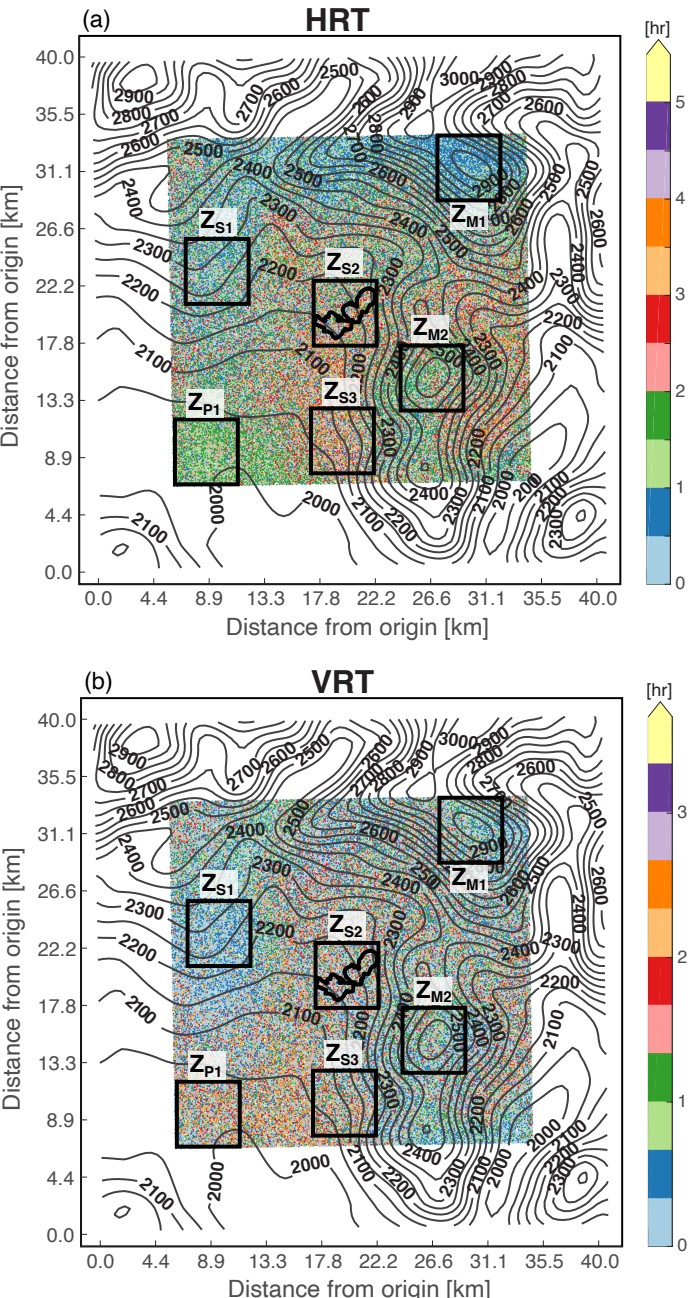

**Figure 7.** Initial particle positions within release zone, with marker color indicating (**a**) horizontal residence time (RT) (HRT) (h) within a 5-km-radius circle centered on the particle release point and (**b**) vertical RT (VRT) (h) within a 250-m-deep layer above the particle release level. Contours of surface elevation (dark gray lines; interval: 50 m) are overlaid on the particles, the black polygons in the center of each panel depict the perimeters of the two burn units active on 10 September 2018, and the six analysis zones discussed in the text are indicated with black squares: $Z_{M1}$, $Z_{M2}$, $Z_{S1}$, $Z_{S2}$, $Z_{S3}$, and $Z_{P1}$ (see text for naming convention).

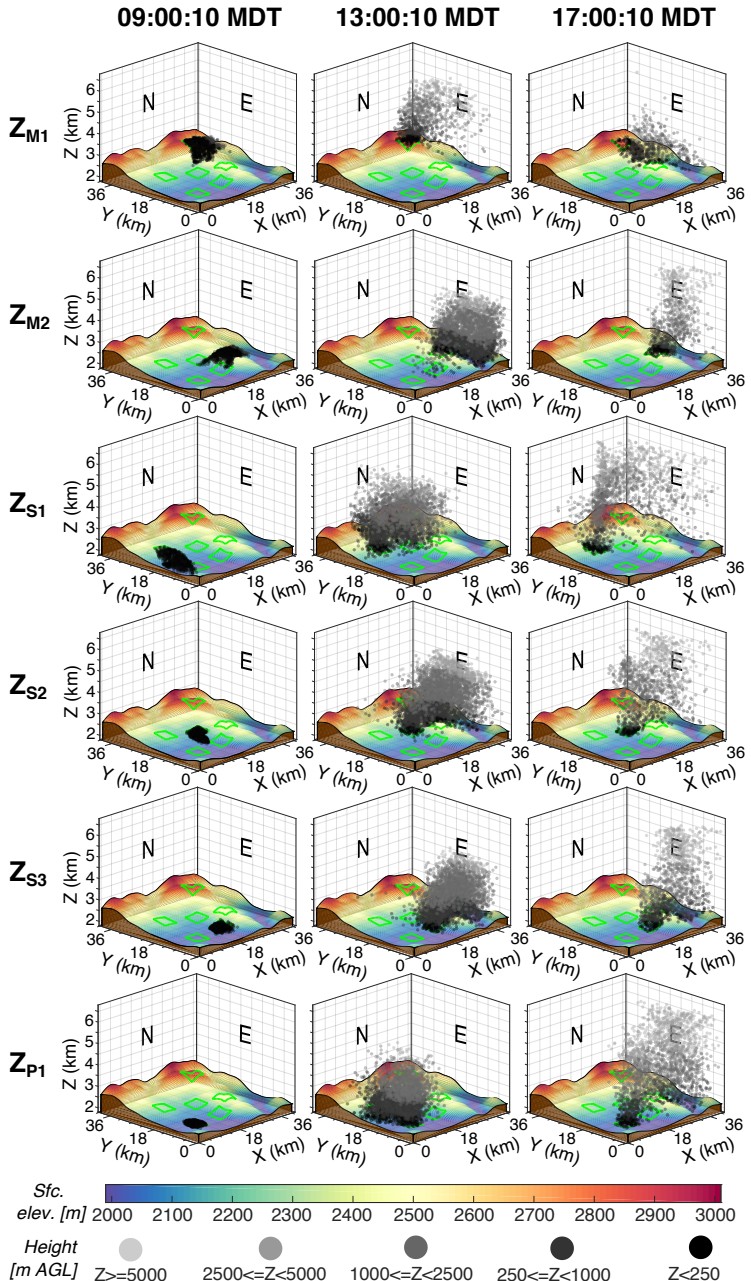

**Figure 8.** Visualization of six PM2.5 "plumes" created by isolating particles released from each analysis zone, from a viewing position southwest of the plumes, with zones organized by terrain characteristics (Mountain, Slope, Plains). Surface elevation (m above mean sea level) is indicated by shading of the three-dimensional ground surface, and particle height (m AGL) is indicated by marker color; see legends at bottom of figure. Analysis zones are indicated with green polygons. Three times are displayed, spanning the period from (**left**) 09:00:10 MDT to (**right**) 17:00:00 MDT, with a 4-h interval between columns.

### 3.4. Ensemble Mean RT Analysis

Next, the strength of the relationship between VI and dispersion is examined via scatter plots of mean VI and ensemble mean HRT (Figure 9) and VRT (Figure S1; supplementary material); the methodology summarized here is identical to that of [8]. For each analysis zone (Figure 7) and for each 10-min time block, mean VI is computed by spatially averaging VI across the zone, and temporally averaging the resulting quantity over the 10 values within the time block (1-min ARPS

output). Ensemble mean RT is computed by identifying the particles released within the analysis zone during the time block, and averaging RT for that subset of particles. For an ensemble mean to be computed for a given analysis zone and time block, a minimum of 100 particles released during that block must have reached the 5-km radius (HRT) or 250-m depth (VRT); otherwise, the time block is omitted from the VI-RT analysis. The outcome of this procedure is a series of scatter plots for mean VI and ensemble mean RT; the process yields 60–65 (52–60) data points per panel for HRT (VRT). As a method of assessing the strength of the VI–RT relationship, linear regression lines are fit to the data points, and corresponding statistics are computed: normalized slope (S), computed as the slope divided by the plot aspect ratio $[(y_{max} - y_{min})/(x_{max} - x_{min})]$, and coefficient of determination ($R^2$). Before proceeding, it is important to clarify our rationale in choosing a linear function for this assessment of VI diagnostic potential, rather than a non-linear function that is a better fit to the scatter plot points (e.g., exponential). We wish to compare how the linear regression, a straightforward measure of how RT (i.e., dispersion) varies with changes in VI, differs between analysis zones. Also, use of the linear regression allows for a more direct comparison of the statistics between this study and [8]. We are not attempting to develop an empirical model for predicting RT based on a particular value of VI.

Comparing the VI–HRT scatter plots and linear regression lines in Figure 9, the analysis zones can be divided into two groups based on $R^2$: zones with $R^2$ less than 0.5 ($Z_{M1}$, $Z_{S1}$, and $Z_{S2}$), and zones with $R^2$ greater than or equal to 0.5 ($Z_{M2}$, $Z_{S3}$, and $Z_{P1}$). In the first (second) group, the linear regression explains less (more) than half of the variability in HRT. It is worth noting that the robustness of the VI–HRT relationship is independent of the general analysis zone grouping (mountain, slope, plains). It is also worth noting that the $R^2$ and S values in the second group (0.59–0.72 and $-0.31-{-}0.58$, respectively) are smaller than those judged robust in [8] (0.79–0.8 and $-0.63-{-}0.67$, respectively). The smaller values of $R^2$ and S noted in this study have two likely sources. First, the occurrence of a complete daytime PBL cycle during the Saul's Creek event, with a stable PBL transitioning to a convective PBL and then back again, means that particle dispersion is simulated during both poorly-mixed and well-mixed atmospheric states (unlike the exclusively well-mixed state in [8]); note in Figure 9 that HRT sharply decreases with increasing VI in the extreme left side of the scatter plots. Second, the occurrence during the Saul's Creek event of thunderstorm outflows yields points on the scatterplot distant from the linear regression line (see markers with colors corresponding to release times of 17:00–19:00 MDT). For these particle bins, VI is small because MH is small (due to the stably stratified air within the outflows), but horizontal dispersion is strong due to the enhanced wind speeds within the outflows. Finally, maximum HRT in the upper-left corner of each panel ranges from 3–3.5 h for $Z_{S2}$ and $Z_{S3}$, to 2 h for $Z_{M2}$, $Z_{S1}$, and $Z_{P1}$, to 1 h for $Z_{M1}$. The sharp decrease in HRT with increasing VI is limited to VI values of approximately $0.2 \times 10^4$ m ms$^{-1}$ or less; for VI greater than $0.2 \times 10^4$ m ms$^{-1}$, HRT decreases with increasing VI at about the same rate in all zones ($1 \text{ h}/1.8 \times 10^4$ m ms$^{-1}$). Put another way, HRT varies by about one hour across the range of Colorado VAR categories from "Poor" to "Very Good". Lastly, $R^2$ and S values are consistently smaller for VRT than HRT (cf. Figures S1 and Figure 9), with changes to VRT with increasing VI limited to the far left edge of the VI-VRT scatter plots. Thus, VI appears to have less diagnostic potential for vertical dispersion than for horizontal dispersion, consistent with [8].

Having established the relative strength of the VI–HRT relationship across the six analysis zones, the desire for insight into the underlying mixing and transport processes motivates an assessment of the relationship of HRT to the individual components of VI (MH and $WS_{ML}$), as well as WS averaged across alternative depths (Figure 10). This is a similar approach to [8] (see their Figure 7), however, in this study the occurrence of a full diurnal cycle and the passage of thunderstorm outflows through the FLEXPART-WRF domain necessitates the addition of a time component to the scatter plot figure (see marker color in Figure 10). For convenience, the VI–HRT scatter plots from Figure 9 are reproduced in the first column of Figure 10, but with uniform marker size (no HRT standard deviation). Broadly speaking, the first three columns of Figure 10 reveal a decreasing trend of HRT with increases in VI and

its components, MH and $WS_{ML}$. Across the six zones, mean MH ($WS_{ML}$) increases from approximately 0 km (0 ms$^{-1}$) before 09:00 MDT to 3–4 km (5 ms$^{-1}$) by 15:00 MDT, with HRT exhibiting a decreasing trend during this period (albeit with considerably different slopes). Such a general finding is consistent with the underlying principle behind VI, that is, that smoke dispersion increases with increasing mixed layer depth and transport wind speed. Additional details of the diurnal cycle of VI, MH, and $WS_{ML}$ are provided in Figure S2 (supplemental material).

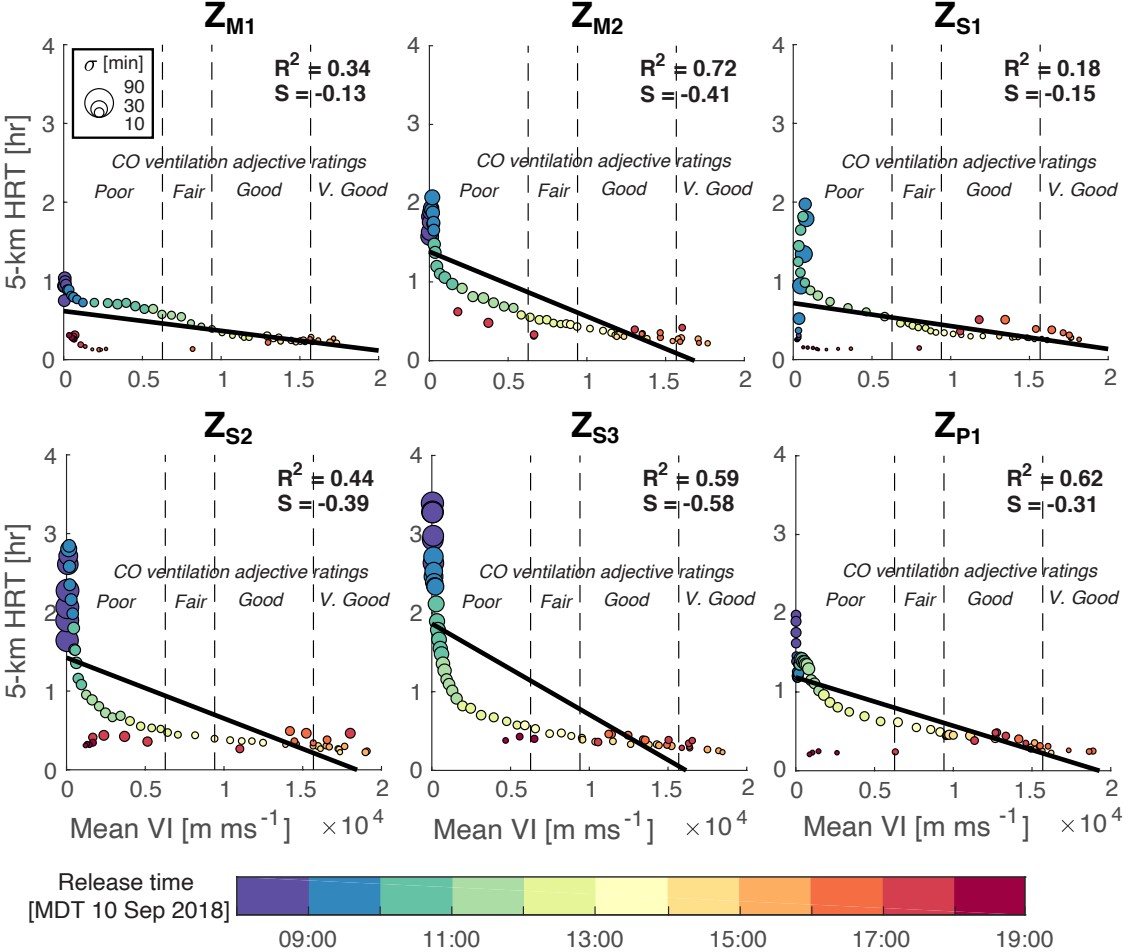

**Figure 9.** Scatterplots of 10-min-mean, zone-averaged ventilation index (VI) vs. ensemble-mean HRT, for each analysis zone. The size of each circle is proportional to the standard deviation of HRT within that time bin (see legend in top-left panel), and the marker color corresponds to the one-hour release time block indicated in the color bar below the figure. The thick black line is the linear regression, S is the normalized slope, computed as the slope divided by the plot aspect ratio [($y_{max} - y_{min}$)/($x_{max} - x_{min}$)], and $R^2$ is the coefficient of determination. State of Colorado VI adjective ratings are provided for reference only (adjective ratings developed by the state of Colorado, with little documentation of their origin).

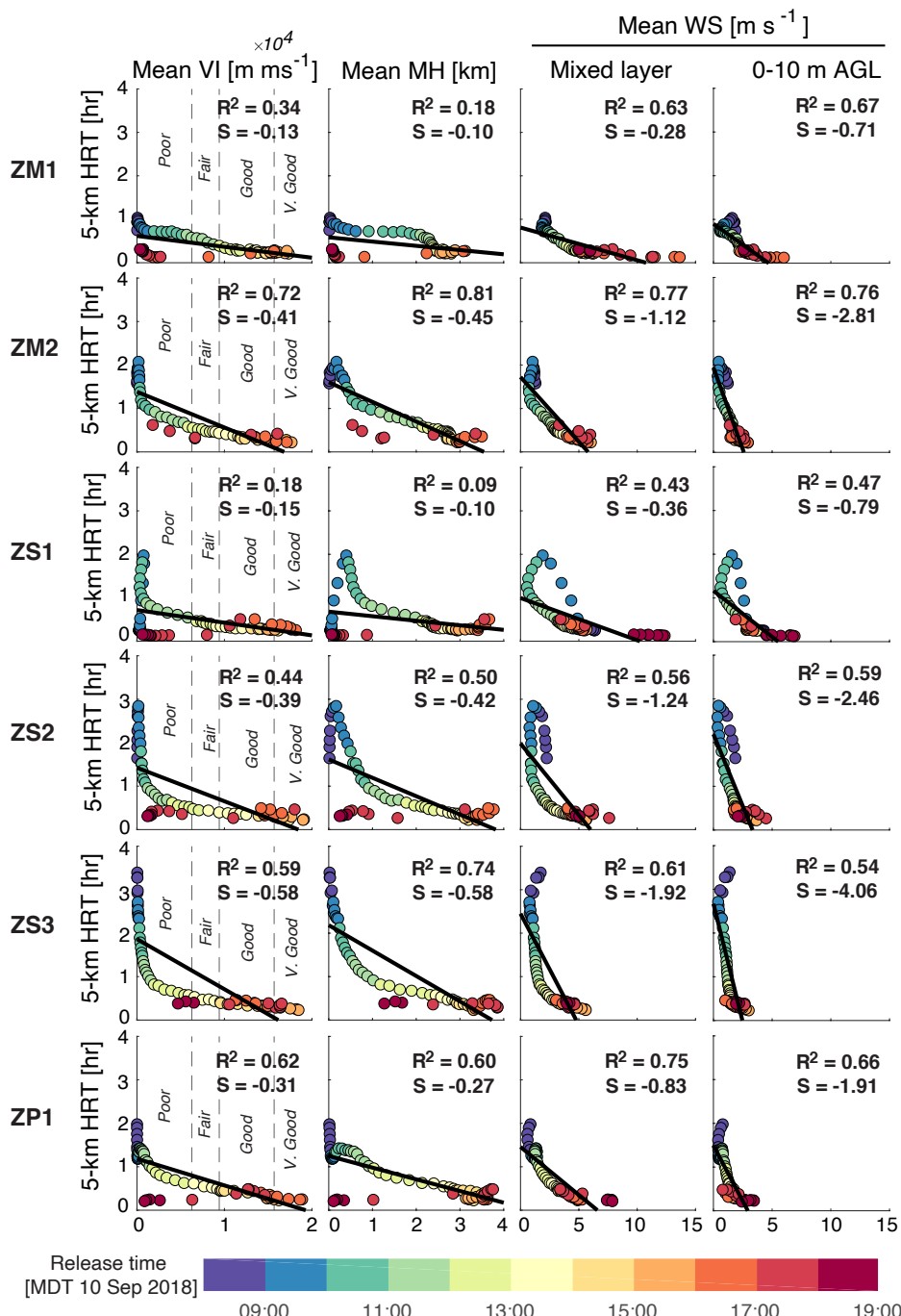

**Figure 10.** Scatter plots of ensemble mean HRT (*y*-axis), and (from left to right) mean VI, MH, and WS (mixed layer and 0–10 m AGL) (*x*-axis). Marker color corresponds to the one-hour release time block indicated in the color bar below the figure. The thick black line is the linear regression, S is the normalized slope, computed as the slope divided by the plot aspect ratio [$(y_{max} - y_{min})/(x_{max} - x_{min})$], and $R^2$ is the coefficient of determination. State of Colorado VI adjective ratings are provided in the first column, for reference only (adjective ratings developed by state of Colorado, with little documentation of their origin).

However, there are two details in the MH and $WS_{ML}$ scatter plots that merit further analysis. First, in $Z_{S1}$ and $Z_{S2}$, the period from 08:00–10:00 MDT is characterized by an increase in MH, and, counter-intuitively, an increase in HRT. The coincident increase in MH and HRT is associated with a decrease in $WS_{ML}$, which based on the location of the analysis zones along north to northeast facing

slopes (Figure 7), suggests that the growth of the mixed layer is serving to weaken downslope flows and thus dispersion. This phenomenon occurs to a lesser extent in $Z_{M1}$ and $Z_{M2}$, but is absent from $Z_{S3}$ and $Z_{P1}$. Following this brief episode, increasing MH and $WS_{ML}$ are associated with decreasing HRT. Second, the points displaced toward the lower-left corners of the VI–HRT scatter plots, identified in the discussion of Figure 9 as being related to the thunderstorm outflows, are also evident in the MH panels. As the outflows impinge on the analysis zones, MH decreases by 50% or more from pre-outflow values, reducing VI from values identified by the Colorado VAR system as indicating "Good"/"Very Good" dispersion potential to values indicating "Fair"/"Poor" dispersion potential, despite the increase in $WS_{ML}$ associated with the outflows; however, despite the dramatic reduction in VI following the thunderstorm outflow passage, HRT is either unchanged or smaller than the pre-outflow afternoon values. This illustrates an important limitation of VI for indicating dispersion potential: in situations wherein horizontal dispersion is primarily driven by near-surface winds in a stably stratified layer, a MH-dependent diagnostic tool such as VI fails.

Finally, the relationship between HRT and wind speed vertically averaged from 0–10 m AGL ($WS_{10}$) is examined (Figure 10, last column). The authors of [8] found that in locations where VI performed least well (e.g., lee of terrain obstructions), shallow layer wind averages had greater diagnostic value than VI or its components. Comparing $R^2$ and S across the four columns, it appears that in the analysis zones where VI performed least well ($Z_{M1}$, $Z_{S1}$, and $Z_{S2}$), the statistical measures of diagnostic value are indeed greater for $WS_{10}$ than the other three variables (VI, MH, $WS_{ML}$). The location of these three zones in the northern half of the FLEXPART-WRF domain (Figure 7), closest to the O(3000 m) terrain to the north, lends support to the utility of shallow layer wind averages in areas that are most susceptible to the influences of steep terrain on PBL flows.

## 4. Conclusions

In this study, a Lagrangian particle dispersion model (FLEXPART-WRF) was utilized to simulate smoke dispersion in complex terrain in southwestern Colorado. The focus of the study was a prescribed fire conducted from 8–12 September 2018 in the Saul's Creek area of the San Juan National Forest in southwestern Colorado. Model simulations of the background meteorology and smoke plume behavior on 10 September 2018 (primary burn day) were utilized in this study to address two research questions. First, is a horizontal grid spacing of 4 km sufficient to capture the spatiotemporal variability in wind and PBL structure during the Saul's Creek fire, or is finer grid spacing necessary? Second, what is the relationship between VI and smoke dispersion during the Saul's Creek fire event, as measured by particle RT within a given horizontal or vertical distance from each particle's release point, and are the current Colorado VAR categories an adequate indication of dispersion conditions?

First, the ability of the ARPS model to characterize the background meteorological conditions was evaluated using a suite of observations, both external and internal to the prescribed fire experiment. The ability of the model to capture the vertical variations in wind speed and direction during the morning hours was sensitive to grid spacing, with domains D2 (1.33-km horizontal grid spacing) and D3 (444-m horizontal grid spacing) comparing better with the observed profile than domain D1 (4-km horizontal grid spacing). During the afternoon hours, differences between the three model domains were more subtle and all three domains captured the observed vertical structure similarly; however, the timing of the thunderstorm outflow passage was sensitive to model grid spacing, with the arrival of the temperature drop and wind shift differing by two hours between domain D1 and domains D2 and D3. Also, peak mixed layer heights were approximately 1 km greater in domains D2 and D3 than in domain D1, likely the result of partially-resolved PBL processes in the inner domains. Taken as a whole, the ARPS validation exercise suggests that horizontal grid spacing of approximately 1 km or less is needed to capture the spatiotemporal variations in wind and PBL structure during the Saul's Creek fire.

Considerable differences in horizontal RT (i.e., dispersion) at different elevations were identified, with VI exhibiting greatest diagnostic value in the southern half of the domain, and lowest diagnostic

value in the northern half of the domain, closest to areas of higher terrain with surface elevations O(3000 m). Further analysis of VI and its components (MH and $WS_{ML}$) revealed two caveats to the broad conclusions regarding VI diagnostic value. First, outside of VI values below about $0.2 \times 10^4$ m ms$^{-1}$ (low end of the "Poor" Colorado VAR category), HRT steadily decreased by about one hour as VI increased across the range of Colorado VAR categories from "Poor" to "Very Good". Thus, this study was unable to corroborate the Colorado VAR category thresholds as breakpoints for dispersion regimes. Instead the VI–HRT scatter plots suggest two distinct dispersion regimes, one with VI values below about $0.2 \times 10^4$ m ms$^{-1}$, in which HRT decreases by 1–3 h across the narrow range of VI values, and a second regime with VI values in the broad range of $0.2 \times 10^4$–$2.0 \times 10^4$ m ms$^{-1}$, in which HRT varies by less than 1 h. Second, despite the dramatic reduction in VI following the passage of thunderstorm outflows, HRT was either unchanged or smaller than the pre-outflow afternoon values. Put simply, the fundamental VI assumptions (horizontal homogeneity, stationarity of mixed-layer depth and wind speed) are violated during thunderstorm outflow passages.

To summarize, the results of this study confirm in part the finding of [8] that VI has some diagnostic value in areas of complex terrain. The extension of the [8] framework from eastern Pennsylvania (O(400 m) elevation change from ridge to valley) to southwestern Colorado (O(1000 m) elevation change from ridge to valley) further increases confidence in this general conclusion; however, the presence in the current study of a full diurnal cycle and a thunderstorm outflow passage have allowed for a more complete evaluation of VI. The absence in this study of a relationship between the Colorado VAR category breakpoints and HRT suggests that future work may be needed to establish breakpoints with a more physical basis. The breakdown of the fundamental assumptions underpinning VI during and after thunderstorm outflow passage suggests that land managers and other VI users also take into account forecasts of thunderstorm potential when making burn decisions. Finally, based on the multiple domain assessment of wind and PBL structure in this study, it is recommended that fire weather and smoke forecasts in areas of complex terrain be based on forecasts from numerical models with horizontal grid spacing of 1 km or less, when possible.

The application of the modeling framework used in this and the previous study to additional prescribed fires in Colorado is expected to further elucidate the strengths and limitations of VI as a diagnostic tool for evaluating smoke dispersion potential. An unexplored factor that will be addressed in these future studies is the sensitivity of VI diagnostic value to plume rise; in other words, how does VI diagnostic value change when the impact of the fire on plume buoyancy is taken into account? Additionally, the sensitivity of VI diagnostic value to the method used to compute mixing height will be examined in future work. Although this study utilized the surface virtual potential temperature method used operationally by the NWS, the authors of [45] evaluated alternative methods and found that a TKE-based mixing height calculation compared most favorably with LiDAR-estimated mixed layer heights. Taken as a whole, the results from this series of numerical model experiments are expected to help guide the application of VI in complex terrain, and possibly inform development of new metrics for dispersion potential.

**Supplementary Materials:** The following are available online at http://www.mdpi.com/2073-4433/11/8/846/s1.

**Author Contributions:** Conceptualization, M.T.K., J.J.C. and T.O.M.; data curation, M.T.K. and X.B.; formal analysis, M.T.K., J.J.C., S.Z., W.E.H., and X.B.; funding acquisition, J.J.C. and T.O.M.; investigation, M.T.K., J.J.C., S.Z., W.E.H., and X.B.; methodology, M.T.K., J.J.C., and S.Z.; project administration, J.J.C., S.Z., and T.O.M.; resources, T.O.M.; validation, M.T.K.; visualization, M.T.K.; writing—original draft, M.T.K.; writing—review and editing, M.T.K., J.J.C., S.Z., W.E.H., X.B., and T.O.M. All authors have read and agreed to the published version of the manuscript.

**Funding:** This research was funded by the United States Department of Agriculture (USDA) Forest Service via Research Joint Venture Agreement 17-JV-11242306-042. This work was supported partially by the USDA National Institute of Food and Agriculture, Hatch project 1010691.

**Acknowledgments:** We wish to thank the Columbine Wildland Fire Module for conducting and managing the prescribed fires for our experiments. The color map used in all figures except Figures 2 and 4 was developed by Cynthia Brewer at the Pennsylvania State University [url: http://colorbrewer2.org/]. Finally, comments and suggestions from three anonymous reviewers were helpful in revising the manuscript and are greatly appreciated.

**Conflicts of Interest:** The authors declare no conflict of interest. The funders had no role in the design of the study; in the collection, analyses, or interpretation of data; in the writing of the manuscript, or in the decision to publish the results.

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
