# Peer review of "A Multiscale Numerical Modeling Study of Smoke Dispersion and the Ventilation Index in Southwestern Colorado"

_atmosphere, doi:10.3390/atmos11080846_

Round 1
Reviewer 1 Report
Overall, I believe this study is interesting and demonstrates a creative approach (with the use of FLEXPART-WRF) for evaluating the diagnostic value of the ventilation index in complex terrain to inform on dispersion metrics used in smoke management applications related prescribed fire. The writing is clear and well organized. This manuscript should be accepted, but I would ask that the authors please consider and address the following major and minor comments below.
Major Comments:
Comment 1: Related to mixing height (MH) and the application of the Stull method; adding a perturbation (an increase) of 1K to the surface parcel for the reasons specified—to bypass weak inversions in the model. The original intent of adding 0.5K to the surface parcel was to create a minor offset (or a very small perturbation) relative to the environmental sounding in order to evaluate surface parcel buoyancy. This particular application of the Stull method was well tested across the conterminous US in ref [42]. Of course, the use of 0.5K is imperfect and an argument can certainly be made that 1K or some other factor improves surface buoyancy and MH determination. However, I'm not fully convinced that changing the 0.5K perturbation used currently by the NWS is appropriate or addressing the underlying issue, ie, weak inversions above the surface limiting the vertical extent of surface parcel buoyancy and producing MH underestimation. The cold surface bias in the model stands out more as the underlying source problem, as does the 5 m agl model surface layer in D3 (Fig. 5 caption), which may be too high above agl yielding a cooler surface parcel. Figure 3 shows the superabiabatic surface layer in the rawinsonde profile that is either absent or considerably weaker in D3—a zoom of these lowest layers would be diagnostically informative. Also, in Figure 4, the rawinsonde MH appears quite a bit higher than the ceil and model with time (the absence of the blue triangle after 1400 MDT is weird too). And I believe in this study, the 1K factor applied is universally to all surface parcel thetaV when estimating MH, is that right? I believe adding 1K to the surface parcel in the rawinsonde profile (with the superadiabatic layer) erroneously inflates the MH; by how much is unclear. A bias correction to the model surface might be a more effective approach, although I don’t advocate doing that at this stage. I don’t think use of the Stull method in this way (adding 1K to the surface) comprises the conclusions, but I do think these points should be clarified a bit (e.g., a few short statements or figure in supplemental form) and caution should be used when adding additional heat to the surface parcel for use with any parcel-MH method.
Comment 2: The Stull method was chosen for MH estimation in this study to be consistent with NWS, is that right? If so, I understand that aspect. However, it seems the VI and dispersion comparison and conclusions could be further supported with TKE MH determination given the availability of prognostic TKE in the numerical model. I understand TKE does not translate to the rawinsonde MH estimation. It is comparable to the ceil Lidar. I suspect TKE MH would better capture the impact of the thunderstorm outflow and, in general, gradients associated with dynamics and thermodynamics occurring within complex terrain. I wonder how TKE would translate in the VI calculation; could this be tested or commented on as a supplemental for one of the analysis zones?
Minor comments:
Figure 4: The rawinsonde MH, blue triangle, is absent after 1400 MDT; just curious why that is?
Figure 5 caption: Lowest model grid levels are discussed against observation height. It would be helpful to have these numbers for each domain listed in Table 1 as well.
Reviewer 2 Report
Major Comments:
There is no verification that modeled residence times represent the true dispersion in the mixing layer. Since the smoke concentration data could not be used for this purpose, there should be at least a demonstration of convergence of numerical model results. This requirement is elaborated in the following two comments.
1) The meteorological model performance is evaluated for different grid resolutions. While it is important to know how well the PBL structure and wind fields are captured with different grid resolutions, this information is not sufficient for judging how grid resolution affects VI's representativeness of particle residence times. For example, would you arrive at the same conclusions if you restricted the analysis in Section 3.4 to D2? Or, if you expanded the analysis to finer grid resolution, say to a D4, would you see additional limitations of VI? How do you know whether the grid resolution is sufficient?
2) How sensitive are the results in Section 3.4 to the selection of horizontal and vertical length scales for residence time calculations (i.e., 5 km and 250 m, respectively)? For example, what would happen if you used the values in [8], which were smaller. Or, if you used larger values, would you still arrive at the same conclusions?
Minor Comments:
1) A discussion of the three aerosol layers in Figure 4 would be useful.
2) A reminder of the horizontal grid resolutions for D1-D3 in the Conclusions would be useful.
Reviewer 3 Report
The paper deals with the estimation of ventilation index in a complex terrain during fire and smoke dispersion. The subject with its relation to wildland fires is very important especially nowadays. However I find the paper interesting and important, the presentation is not fully clear, and some parts should be clarified or supplemented. Some general remarks:
- The sentences in the text are sometimes very long which make the text in some parts difficult to understand. Especially for not native speakers, like me. Some of my remarks can be caused by possible misunderstanding of some parts.
- The Authors gave the detailed description of obtained results. I suggest to try to formulate more specific analyses and conclusions. The Authors wrote (l.550) ”… VI has some diagnostic value in areas of complex terrain.” As I understand because it indicates the concentrations of particles. So, why not use concentration as the indicator?
- ARPS for weather data and FLEXPART-WRF for particle dispersion are used. Please, give some reasons why these were used? There are many tools for meso-scale weather forecasts/simulation and also for fire or smoke dispersion. Please give the advantages of used tools.
As I understand the Authors used ARPS for wind simulation and then in the properly simulated wind field – FLEXPART to simulate dispersion? As I understand you simulate wind according to available environmental data. What exactly were boundary conditions for that simulation? Then you compared the obtained results with measurements made in various points inside domain. So this can be validation of the wind simulation. After that, the Authors started Flexpart and simulated dispersion in the obtained wind field? And these were boundary conditions for Flexpart? I do not see the real fire/smoke simulation? In my opinion, You can simulate dispersion of smoke from the given point and compare results with the real smoke plume. There is no such comparison here. The whole procedure is not clear for me.
Please clarify (e.g. end of introduction) what was simulated, on which bases, what results were compared as validation. - What does Dzmin in Table 1 mean? Is this vertical grid dimension at ground level. And why is it called "nested" here? Do you mean D1-D3 nested in the bigger area?
- In general, what was the purpose of using 3 domains D1-D3. In my opinion the statement that if we use more fine mesh we will get better results of simulations is evident. Especially when talking about wind speed and direction. There are too many factors associated with terrain
topography and orography and ground coverage influencing the flow.
I understand that this is one of the approaches to the problem. Please comment, if possible, why you didn’t place D3 in D2 and both in D1 by increasing the resolution or D3 directly in D1. The desired area D3 with fine grid could be nested in the big area D1 with coarse grid? In such case the problem of re-scaling wind data from meso-scale to micro scale appears.
Please comment, if CFD RANS or LES approaches can be used for such purposes?
This question is connected with my opinion, that in all such approaches, the resolution near the ground is not sufficient for the real representation of the Atmospheric Boundary Layer. And ABL are crucial for spread of the fire and smoke. Of course we have to find the compromise between accuracy of simulation and its time and effectiveness.
Moreover, using such resolution, information about topography and orography of the terrain can be omitted and thus propagation of wind can be burden with error.
It is just a remark of discussing character because it depend on the approach to the problem. - What was the reason to adopt 5 km radius circle centered on the particle released point.
Perhaps I didn’t understand but the release was uniform in the domain of 28x28 m2 (line 234). The coupled action of wind and temperature just produced the movement of the particles in the whole domain? If so, what is the relation to the described fire? I think, that the smoke release should be placed in 10 and 11? How does the whole simulation correspond to the real event (Fig.8 – 6 zones, and only zS2 is in the center of fire?). - It is difficult to analyse the wind direction and speed without more detailed information of the terrain roughness. Give more details about it, perhaps map with the terrain.
- What was the height for wind speed and direction measurements (and temperature also)(Fig. 2)? The values of wind speed and direction in Figs 2 and 3 – are they mean or instantaneous? Perhaps provide some more information about the measurement procedure. Line 295 – S, VH here are wind speed and direction, respectively. Consider using V for wind speed as usually.
- line 194 – what is "virtual potential temperature"?
- Fig. 3. I think that more information about the grid close to the ground is necessary. The major changes in wind direction and speed are close to the ground. Please give description of the axes in fig. 2.
- Fig. 6. What is 10 m/s in all figs? And how does it correspond to Fig. 3, and especially to Fig. 2 (where maximum speed is 35 m/s on the wind rose)?
- How is VI calculated? Fig. 8. (see also above comment) – did you take the particles from the given zone and analyse only their dispersion? So, how does it correspond to the real event? What is mean VI? Figs 9 and 10 – there is division into "poor, fair, good ,very good". What are the bases for such division?
- I suggest to add supplemental Figures to the text.
- Line 498 – what do you mean by "wind speed averaged from 0-10 m"?
- More details about thunderstorm are needed as the Authors make the analysis during such event.
Round 2
Reviewer 1 Report
I appreciate the careful consideration given to the major comments and the responses provided. I also believe the revisions help address some key aspects and caveats of the mixing height calculation used in this study. I also agree that a future study evaluating VI with multiple mixing height methods would be interesting, particularly for a study area with complex terrain.
Reviewer 2 Report
Thank you for considering and responding to my comments.
Reviewer 3 Report
Thank you for your answers. I analysed my previous remark 7 again and I can agree with you.
No more questions.